TOPICAL REVIEW

# An update on pacemaking in the myometrium

Susan Wray[1] and Michael J. Taggart[2]

[1] *Women's & Children's Health, Faculty of Health & Life Sciences, University of Liverpool, Liverpool, Merseyside, UK*
[2] *Biosciences Institute, International Centre for Life, Newcastle University, Newcastle, UK*

Handling Editors: Laura Bennet & Bernard Thomas Drumm

The peer review history is available in the Supporting Information section of this article (https://doi.org/10.1113/JP284753#support-information-section).

**Abstract**   Timely and efficient contractions of the smooth muscle of the uterus – the myometrium – are crucial to a successful pregnancy outcome. These episodic contractions are regulated by spontaneous action potentials changing cell and tissue electrical excitability. In this short review we will document and discuss current knowledge of these processes. Those seeking a conclusive account of myometrial pacemaking mechanisms, or indeed a definitive description of the anatomical site of uterine pacemaking, may be disappointed. Rather, after almost a century of investigation, and in spite of promising studies in the last decade or so, there remain many gaps

**Susan Wray** is Emeritus Professor of Molecular and Cellular Physiology at the University of Liverpool. She is internationally-renowned for her research in pregnancy and women's health especially her contributions to understanding the mechanisms of uterine smooth muscle excitation-contraction coupling in relation to normal and dysfunctional labour. She is a fellow of the Academy of Medical Sciences and of the Royal College of Obstetrics and Gynaecology. She has been a former editor of the Journal of Physiology, inaugural Editor-in-Chief of Physiological Reports and is the current President of the International Union of Physiological Sciences. **Michael Taggart** is Professor of Reproductive Sciences at Newcastle University. His research has focused on studying the molecular, cellular, tissue and organ remodelling events that occur in the mother and fetus during pregnancy – including in pregnancies complicated by the onset of spontaneous premature birth – and the impact these have for postnatal life to adulthood. This work encompasses investigation of uterine, vascular and cardiac physiology. He is a Fellow of the Physiological Society.

The Journal of Physiology

in our knowledge. We review the progress that has been made using recent technologies including *in vivo* and *ex vivo* imaging and electrophysiology and computational modelling, taking evidence from studies of animal and human myometrium, with particular emphasis on what may occur in the latter. We have prioritized physiological studies that bring us closer to understanding function. From our analyses we suggest that in human myometrium there is no fixed pacemaking site, but rather mobile, initiation sites produce the connectivity for synchronizing electrical and contractile activity. We call for more studies and funding, as physiological understanding of pacemaking gives hope to being better able to treat clinical conditions such as preterm and dysfunctional labours.

(Received 11 December 2023; accepted after revision 24 April 2024; first published online 29 July 2024)

**Corresponding author** S. Wray: Women's & Children's Health, Faculty of Health & Life Sciences, University of Liverpool, Liverpool, Merseyside, UK.     Email: s.wray@liv.ac.uk

**Abstract figure legend** The spread of multiple electrical signals (panel *A*, blue-to-red indicates increasing electrical excitability) that are spatiotemporally distinct, yet in-phase with the excitatory episode, determines action potential shape and form (panel *B*, as recorded by single cell microelectrodes) and ensures contractile amplitude and duration (panel *C*). Time scale bar (appropriate for humans and guinea-pigs) 200 m s.

## Introduction

The uteri of all species produce repetitive coordinated contractions at term with the purpose of safe delivery of their young and placentas. Periodic alterations in uterine smooth muscle electrical activity underlies these contractile events. The regulator and source of this electrical rhythmicity, a uterine pacemaker (or pacemakers), has been sought for decades. The rhythmic coupling of action potential changes and contractions are produced without the need of inputs from nerves or hormones – hence the name spontaneous or myogenic contractions (Wray & Prendergast, 2019). In this short review we consider what evidence there may be for a unique pacemaker(s). We start by describing work looking for the anatomical site of a pacemaker. We do this by describing the structure of the uterus in different species and why this is relevant to pacemaking, and then remark upon studies performed at the organ level using a variety of techniques to find the pacemaker. We then switch from anatomy to functional studies investigating pacemaker mechanisms and uterine myocytes. From this we describe spatiotemporal evidence for electrical pathways in the uterus and consider recent evidence from 3D imaging studies for and against set pacemaking sites. We finish by focusing on computational approaches and assessing how well models derived from them can simulate uterine electrical and mechanical activity, and what that tells us about pacemaking and its mechanisms.

## Pacemaker sites

**Structure and tissues of the uterus.** To understand pacemaking in the uterus requires knowledge of the structure and functions of the organ. While the uterus is responsible for sperm transport, implantation and nurturing and delivering live young across many species, its structural form is markedly varied throughout the mammalian kingdom. In higher primates and women, unless there is an anatomical malformation, the uterus is *simplex*, meaning it forms a single cavity, broader at the top than the bottom, where it is contiguous with a single cervix. In the rest of the animal kingdom, it is divided to a greater or lesser extent, and there may be two cervixes and even two vaginas. For example, pigs, lemurs, dogs, and whales have a *bicornate* uterus, where the upper part is divided into two cavities, often referred to as horns, which transposes to a single cavity at the cervical end. In cats, horses, rodents and ruminants the uterus is divided throughout its length, but both horns join a single cervix, a uterine form known as *bipartite*. Finally, the *duplex* uteri found in rabbits, and marsupials, have duplication of the entire uterus, cervix, and vagina. See Fig. 1 and Machado et al. (2022), for a recent and detailed description of mammalian uterine development. This variety in macroscopic structure while maintaining the same role in producing electrically driven contractions, brings into immediate question the idea of a common location for a pacemaker(s), at least at a gross anatomical level.

The tissues that make up the uterus are consistent across species: serosal perimetrium, the outer connective tissue, which becomes mesometrium as it attaches to the body wall (Wray, 1983), the smooth muscle of the myometrium forming the bulk of the uterus, and the glandular endometrium (or decidua in pregnancy) lining the uterine cavity. The electrically excitable myometrial smooth muscle cells generate the motor force within the uterus, to efface and dilate the cervix, and then propel the fetus(es) and placenta(e) through the birth canal.

**The myometrium.** A cross-section through the uterus reveals its extensive muscular nature. Bundles of myocytes in fascicles of 1−2 mm diameter and several centimetres long, can be readily observed, and dissected for physiological studies (Arrowsmith, 2022). These bundles, which are considered the functional units of the myometrium (Sakai et al., 1992), run in series and in parallel throughout the tissue. The excitability in the bundles needs to be synchronized sufficiently if coordinated contractions are to be produced. As with other excitable cells, electrotonic potentials exist and action potentials are propagated (Garfield et al., 1988; Tomita, 1975), with currents passing through low resistance gap junctions between the uterine myocytes (Garfield et al., 1988). Thinner bundles would be expected to have a lower input current to produce excitation than thicker bundles, assuming a similar degree of connectivity between myocytes, as the thicker bundles have a higher area of cell-to-cell coupling, which is associated with a requirement for larger input currents (Tomita, 1975). As discussed next, modern imaging studies now allow us to measure bundle sizes and reconstruct their orientation and 3D connections.

In a computational study that compared 3D reconstructions with myometrial histological sections, the average muscle bundle widths in non-pregnant human myometrium were similar to pregnant rat myometrium at ∼500 µm (Lutton et al., 2017). It was also found that the bundle width in human myometrium increased from the inner to the outer region. In rat, the bundles were wider in the inner circular than the outer longitudinal layer. These differences may lead to expectations of contraction initiation sites therefore differing between the two species, if thinner bundles are easier to excite.

The classical separation of outer longitudinal fibres orientated along the long axis of the organ, and inner circular concentric layers, found in many visceral smooth muscle tissues, is less evident in the human uterus. It is, however, apparent in other species, including the uteri of rats, mice and guinea-pigs, which are often used for experimental studies. These species allow for investigation of physiological differences between the circular and longitudinal layers. In recent light microscopy studies of rat and mouse, anastomoses and mesh-like connections have been found between the two layers, offering a possibility for electrical and/or mechanical co-ordination between the muscle layers (Kagami et al., 2020).

In the human uterus, (Young & Hession, 1999), the bulk of the fibre bundles of the main muscle mass (sometimes referred to as the outer myometrium) cannot easily be separated into longitudinally and circularly orientated layers (Weiss et al., 2006). This 'outer' myometrium is formed from mesoderm during development, and it is the muscle mass where long-distance electrical communication must occur in support of labour contractions.

MRI and diffusion tensor MRI (DTMRI) imaging of a just-post-pregnant human uterus *ex vivo*, has indicated the complex interweaving of myometrial bundles (Aslanidi et al., 2011). Physical connections between bundles/fasciculi have been described (Young & Hession, 1999), though their molecular and functional characteristics, much like the rodent uteri mesh-like connections mentioned above (Kagami et al., 2020; Lutton et al., 2018), remain to be determined.

The inner myometrium, that is the sub-endometrial, or in pregnancy the sub-decidual, myometrium is a region of more organizational complexity (Naftalin & Jurkovic, 2009): there is no submucosal layer separating the myometrium and endometrium (decidua), the inner myometrium smooth muscle cells are subject to remodelling by invasive placental-derived extravillous trophoblast, and maternal-placental-fetal endocrine, paracrine, metabolic and mechanical stimuli are sensed near this area. Embryologically, the sub-endometrial myometrium and the endometrium share a paramesonephric origin, thereby differing from the outer myometrium. Regulation of the inner myometrium contractility has been suggested to aid sperm transport, implantation, and endometrial

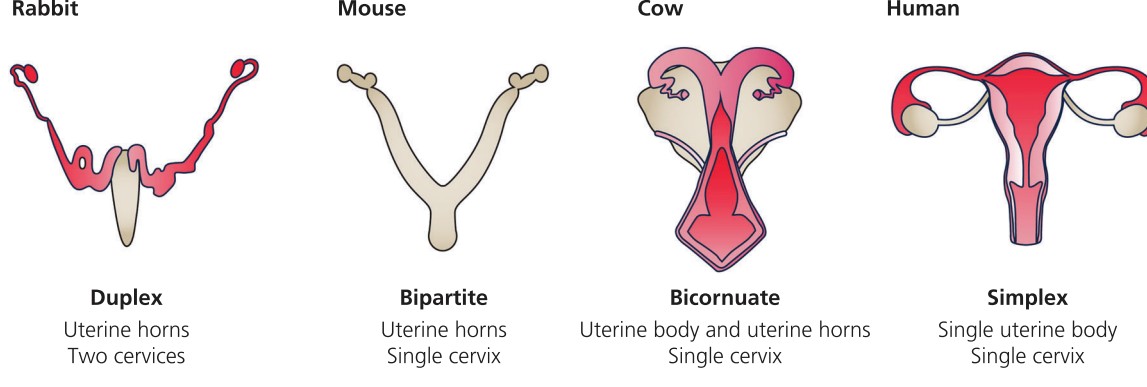

| **Rabbit** | **Mouse** | **Cow** | **Human** |

**Duplex** **Bipartite** **Bicornuate** **Simplex**
Uterine horns Uterine horns Uterine body and uterine horns Single uterine body
Two cervices Single cervix Single cervix Single cervix

**Figure 1. Examples of uterine forms**
The variety of forms the uterus can take throughout the mammalian kingdom.

shedding during menses. Motile activity in this region in non-pregnancy can be seen using transabdominal or vaginal ultrasound. It is reported as chaotic in terms of orientation and frequency and altered by stages of the menstrual cycle, effects attributable to changes in progesterone and oestrogen signalling in this region (Ijland et al., 1996; Kunz et al., 1996; Little et al., 1995). Of interest, motility in this region has been described as less coordinated in non-pregnant women who have a Caesarean section scar (niche seen by ultrasound), compared to those without a scar, (Jordans et al., 2023). There is increasing evidence that emergency C-section late in pregnancy imparts a greater risk of spontaneous preterm birth in each subsequent pregnancy (Suff et al., 2022; Woolner et al., 2024). The focus of attention on underlying reasons has been on damage to cervical tissue at the low site of surgical incision and whether this leads to cervical incompetence in subsequent pregnancies. Quite where the boundary of uterine and cervical (smooth muscle) tissue lies is uncertain. If there is a continuum of smooth muscle tissue it may also be possible (e.g. by analogy to arrythmias in damaged cardiac tissue) that unstable electrical conductivity will ensue. This possibility, and whether it contributes to premature labour, has not been investigated. By analogy to arrythmias in damaged cardiac tissue, this could indicate disrupted pacemaking after Caesarean sections but this does not appear to have been directly studied.

### Electrical activity in myometrium

A full description of the uterus to include its innervation, vasculature and connective tissues is beyond the scope of this review, and other sources may be consulted; for example, Wray and Prendergast (2019) and Wray et al. (2021). Here we focus on uterine myocytes, as single cells or groups in a muscle bundle, to determine their properties before proceeding to describe possible mechanisms of tissue- and organ-level excitation-contraction coupling. In this way, by understanding how electrical activity arises and its functional consequences, we can then discuss these aspects in terms of pacemaking mechanisms.

**Uterine myocytes.** Uterine myocytes conform to the textbook descriptions of smooth muscle cells – they are spindle-shaped, possess a central nucleus, have thick and thin filaments but not organised in a regimented manner, have intracellular organelles, namely the sarcoplasmic reticulum (SR), and mitochondria, and are bounded by a surface membrane replete with ion channels, pumps, transporters, receptors, and invaginations known as caveolae. The cells are connected by gap junctions. There are, however, aspects of uterine myocytes which

distinguish them. First, in the pregnant myometrium, the cells can be enormously long, up to 0.5 mm, making them one the largest cells in the body. Second, as described below, they produce large amounts of inward $Ca^{2+}$ current, the largest described for a smooth muscle at up to ∼1 µA. Third, cross-bridge cycling can produce high amounts of force, which per cross-sectional area, is similar to many striated muscles. And finally, it has a robust $Ca^{2+}$-containing SR permeating throughout the cell from surface to centre (see Fig. 2).

**Uterine interstitial cells.** As described in other articles in this Special Issue, some visceral smooth muscles contain pacemaking cells or networks close to smooth muscle cells, with the interstitial cells of Cajal of the gastrointestinal tract being the best known. Other tissues including some blood vessels, ureter and bladder have, over the last couple of decades, added to the list of smooth muscle-rich tissues beyond the gut with such potential pacemaking mechanisms (Formey et al., 2011; Lang et al., 2006; Monaghan et al., 2012). Morphologically similar stellate cells, sometimes cKit positive, sometimes referred to as telocytes, have been identified in the uterus (Ciontea et al., 2005; Duquette et al., 2005; Peri et al., 2015) but convincing electrophysiological evidence that they are capable of acting as pacemakers is missing or sparse (Duquette et al., 2005; Hutchings et al., 2009). Roles for these cells in, for example, hormone sensing have been postulated (Hutchings et al., 2009).

**Action potentials and ion channels.** In pregnancy the resting membrane potential of myometrial cells has been reported to become depolarised to ∼−55 mV by term (Kuriyama & Suzuki, 1976; Parkington et al, 1999; Pressman et al., 1988). This is in the range akin to that in other

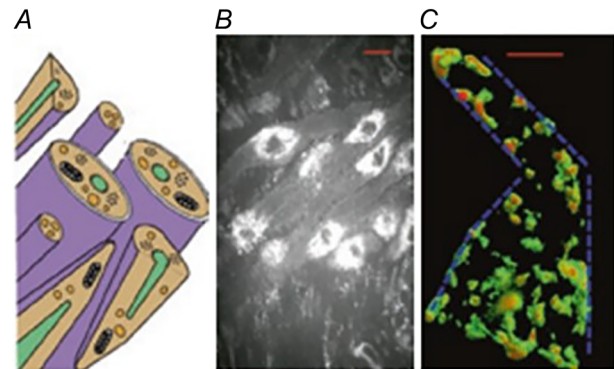

**Figure 2. Uterine smooth muscle cell morphology**
*A*, schematic of uterine smooth muscle cells. *B*, light microscopy of pregnant rat myometrial cells with brightly-stained nuclei. *C*, reticulum distribution in a single uterine smooth muscle cell (blue dotted line indicates boundary of the plasma membrane). Taken from Wray and Prendergast (2019).

tissue systems exhibiting episodic spontaneous activity, perhaps indicative of optimising activation characteristics of inward voltage-gated $Ca^{2+}$ currents for all/most myometrial cells (Shmigol et al., 1998). There are two noteworthy characteristics of myometrial cell action potentials. First, while the action potential form is malleable (Tong et al., 2011), most commonly it involves a sharp upstroke followed by frequent oscillations (spikes) in membrane potential, the repolarising phase of which reaches close to the resting membrane potential. Second, the duration of these action potentials can, depending upon species, last between a few seconds (mouse, rat; Coleman & Parkington, 1992; Taggart et al., 1996) and many tens of seconds (guinea-pig, human; see Fig. 3 and Bengtsson et al., 1984; Parkington et al., 2014). The action potential format, in turn, determines the elicited $Ca^{2+}$ signals and contractile output (Wray et al., 2015).

Many ion channels are expressed in the myometrium and those that play a role in determining the resting membrane potential, and hence the excitability of the myometrium, have been the subject of reviews (Aguilar & Mitchell, 2010; Malik et al., 2021; Wray & Arrowsmith, 2021; Wray et al., 2015). The balance of their contribution is influenced by hormones, uterine environment, and mechanical stimuli (Sanborn, 2000; Shynlova et al., 2009; Taggart & Wray, 1995; Wray et al., 2021). It is now properly appreciated that the changes to ion channel expression evoked by culturing uterine myocytes, especially loss of L-type $Ca^{2+}$ channels, the major contributor to depolarization, renders studies on excitability based on these methodologies inapplicable to the normal physiological processes of excitability and hence pacemaking. Freshly isolated uterine myocytes retain the normal cassette of ion channels and produce electrical and calcium signals when stimulated.

Our increasing knowledge of myometrial cell molecular signatures, and ion channel biophysics, has furnished the development of several mathematical models that quantify the likely contributions of major ion channels and exchangers to action potential-related ion fluxes and fit well to existing experimental data (Atia et al., 2016; Bursztyn et al., 2007; Chkeir et al., 2011; Garrett et al., 2022; Rihana et al., 2009; Tong et al., 2011; Tong et al., 2014; Yochum et al., 2018). Such computational approaches are useful for hypothesis generation and testing about myometrial cell excitability *per se* but also

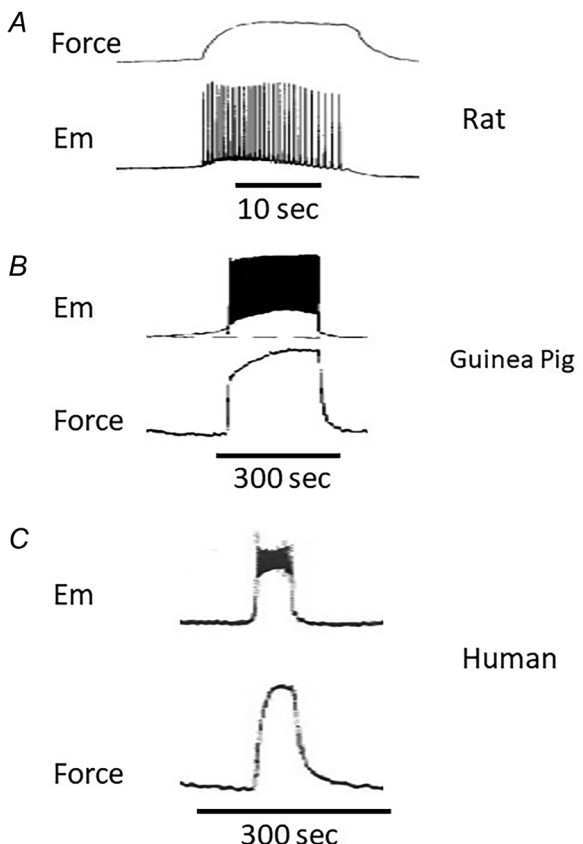

**Figure 3. Uterine action potentials in different species**
Sharp microelectrode recordings of spontaneous myometrial action potentials and accompanying force recordings, from biopsies collected in late pregnancy. Note the different action potential (and contraction) durations between rat (*A*) and guinea-pig (*B*) or human (*C*). From Bengtsson et al. (1984) (*A*), Coleman et al. (2000) (*B*) and Parkington et al. (*C*).

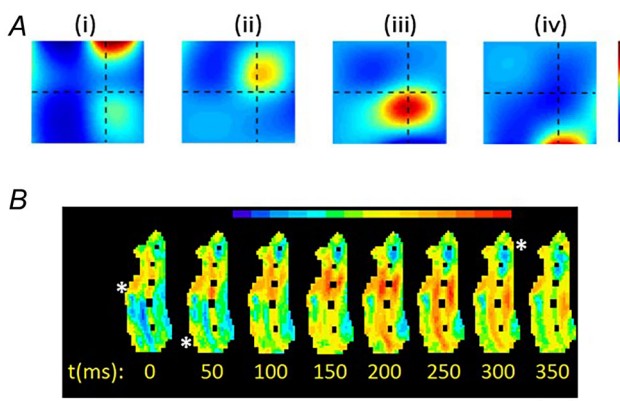

**Figure 4. Spatiotemporal variance in myometrial electrical signals**
*A*, electrohysterogram signals were recorded between 12 evenly spaced electrodes on the abdominal surface (5.25 × 5.25 cm) or pregnant women. Spatiotemporal features of contraction-associated electrical propagations were interpreted by directed information processing. Intense colour clusters in panels (i)–(iv) indicate separate areas of electrical signal termination. From Xu et al. (2022). *B*, optical mapping of voltage-sensitive fluorescence (di-4-Anepps) in isolated guinea-pig myometrial strips during spontaneous electrical activation. Spatially distinct waves of electrical activity can be seen to spread from areas indicated by asterisks at 0, 50 and 300 ms. Scale bar 0.5 cm. From Tong and Taggart (2013).

for assessing the possibilities for pacemaker activity and tissue-level spatiotemporal propagation (see below).

**Spatiotemporal features of uterine electrical excitation.** Pivotal to understanding the mechanisms of excitation-contraction coupling in the uterus is the elucidation of processes underlying initiation and propagation of electrical action potentials. A consistent feature of uterine *ex vivo* electrical recordings over many decades, and in tissues of different species, is that action potentials can be observed to initiate from multiple sites across the organ (Lammers, 2013; Lodge & Sproat, 1981; Parkington, 1985; Young, 2018). Imaging of intracellular calcium signals in myometrium also shows a similar chaotic looking pattern of changes when examined in sheets of myocytes (Burdyga et al., 2007). In uteri from rats close to term, there is an increase in the length constant of conductance thought to, in part, be facilitated by enhanced gap junctional communication (Sims et al., 1982). An important question, however, is in what direction do action potentials travel and how far? In surface multi-electrode array recordings of guinea-pig tissues action potential propagation has proceeded over several centimetres. Evidence of multi-directionality of propagation can be seen in these, and in optical mapping recordings (Lammers, 2008; Lammers et al., 2015; Tong & Taggart, 2013, see Fig. 3), over distances of a few centimetres. In women, the analysis of electrohystogram recordings of human uterine electrical activity from arrays of surface electrodes indicates multi-directional waves of electrical propagation over many centimetres (Lange et al., 2014; Mikkelsen et al., 2013; Xu et al., 2022). Further, *in vivo* magnetomyography recordings of (inter-polated) uterine electrical activity in pregnant women indicates spatially diverse waves of activity spreading over tens of centimetres (Escalona-Vargas et al., 2015; Eswaran et al., 2009; Govindan et al., 2015; Wang et al., 2023). In other papers associated with this Special Issue, the nature of pacemaking in the gastrointestinal tract (GIT) is described. The study of the origins of slow waves in the GIT has been greatly aided by the fine anatomical details of these smooth muscles tissues and the identification of specialized cells, such as ICCs. It is worth noting that when electrical mapping is undertaken, the pacemaking site is seen to shift, "from event to event, even within a small population of muscle cells" (Ward et al., 2006). Descriptions such as these points to some similarity with findings in the uterus. However, unlike the uterus, in the GIT the origin of the pacemaker was not random, and occurred in hotspots, within certain areas of the tissue (Bauer et al., 1985). For a more detailed account, recent papers from the authors can be consulted, along with several comprehensive reviews (Bauer et al., 1985; Huizinga, 2019; Parsons & Huizinga, 2018; Sanders, 2019)

## An anatomically discrete pacemaker?

Ever since anatomical drawings of the human uterus were made by da Vinci (1511) and Vesalius (1543) and their successors, a quest to find the site of the uterine pacemaker began. The identification of the specialized pacemaking region in the heart, the sino-atrial node, gave much encouragement to researchers keen to find the equivalent in the uterus. The data described in the previous section, however, does not lend itself well to theorization of a singular, anatomically distinctive type of pacemaker cell.

In contrast, light has recently been shone on the possible role of the sub-endometrial (decidual) inner myometrium for pacemaking. In pregnant rats a distinct type of myometrial bundle, in between the circular and longitudinal muscle bundles and by the implantation sites, has been postulated to be responsible for initiating bursts of propagating electrical activity (Lutton et al., 2018). Using a combination of electrophysiology, histology and 3D modelling these authors described anatomical areas associated with initiating electrical activity at myo-metrial/placental junctions. Combined with previous data suggesting a dominance of initiating events at the ovarian end of the rat (but not guinea-pig) uterus (Lammers et al., 2015), this work suggests, at the organ level, a possible overall directionality of electrical, and contractile, activity towards the cervix that may benefit evacuation of multiple fetuses and placentas during labour. Specifically, the authors write they saw "distinct structures within the placental bed of individual implantation sites. These pre-viously unidentified structures represent modified smooth muscle bundles that are derived from bridges between the longitudinal and circular laye*rs*". The suggestion is that these myometrial fibres form a conduit between each implanted feto-placental unit and the broader myometrial contractile, smooth muscle network.

The findings of Lutton et al (2018) are to be commended and have served to reignite consideration of the mechanisms of uterine organ excitation-contraction coupling. Several notes of caution about these speculations are appropriate. First, it is possible that tissue damage following removal of the placentas may have predicated a localised region of electrical instability. Second, as indicated above, and accepted by the authors (Lammers et al., 2015; Lutton et al., 2018), initiation of action potential propagations are recorded from elsewhere in the uterus too. Third, and linked, accepting of the possibility of the dominance of electrical signal initiation sites in rat towards the ovarian end and/or maternal-placental zones does not avoid consideration of evidence for multiple sites of action potential initiation, and multidirectional activity, in other species, including guinea-pigs, sheep and humans, where myometrial surface area is markedly larger and the uterine decidua occupied often with only one placental attachment (Parkington et al., 2018). Fourth, the

placental bed zone is an area previously associated with uterine relaxation, as it has high levels of progesterone until close to term (Mesiano et al., 2002; Wilson & Mesiano, 2020). Fifth, a decades-old study compared the features the smooth muscle cells from the placental bed region of rat to non-placental regions (Kanda & Kuriyama, 1980). Throughout gestation the cells from the placental region were significantly more hyperpolarised than those not in this region and conduction velocity was larger, around tenfold, in the non-placental region, again at all stages of gestation. These data are hard to reconcile with a pacemaking site at this region.

The most recent, important novel insight into the activation of term human myometrium is that provided by Wang and colleagues (Wang et al., 2023). Following on from their earlier studies in sheep (Wu et al., 2019), they have now produced non-invasive monitoring of uterine contractions, via electromyometrial imaging, which they call EMMI. Their method can perform *in vivo* 3D, electrical imaging of human uterine contractions during labour, with high spatial and temporal resolution. This allowed characterization of the initiation and dynamics of uterine electrical activation. Of particular relevance to this review was the confirmation of data obtained using older technology, described above (e.g. Garfield & Maner, 2007; Young, 2018), that uterine contractions progress without a predominant activation direction, i.e. they have quite different patterns of activation from one contraction to the next. As the authors write, "Our results indicate that there are no consistent early activation regions in different uterine contractions, and this is direct evidence against an anatomically fixed, cardiac-like pacemaker region in the human myometrium" and long-distance propagation of activation was not seen. This first-of-its-kind study of women in labour has thus provided useful *in vivo* data. It must be pointed out that recordings were only for an hour. In addition, women had an array of up to 192 electrodes attached and were imaged in an MRI machine, all of which is cumbersome and points to the need of further refinements. The prize of obtaining recordings throughout labour, and identifying normal patterns of labour progression *versus* those indicative of pre-term labour, or dysfunctional (slow to progress) labour at term, should encourage further such efforts.

### If not a distinct pacemaker, what?

It is productive to consider how initiating events arising in different areas of the myometrium, should they be instrumental in 'pacemaking', may become ordered enough in time and space over large distances, to produce effective contractions. The resulting increase in intra-uterine pressure would be repeated with each contraction leading to the dilation of the cervix.

The computational experiments of Singh et al. (2012) offered insight to one possible mechanism of long-distance co-ordination of electrical signalling: the emergent co-ordination of previously discrete episodes of oscillating electrical activity by increased gap junctional coupling between heterogenous cells/fasciculi/bundles. This could result in, theoretically, complete synchronisation of electrical activity, should coupling be strong enough. Alternatively, with an intermediate level of coupling, clusters oscillating at slightly different frequencies were proposed. Extending these considerations to incorporate (1) a quantitative cell model of myometrial cell excitability and (2) homo- and heterocellular coupling in a 2-dimensional lattice, Xu et al. (2022) demonstrated a variety of spatiotemporal electrical patterns, again dependent upon coupling strength. These included the generation of non-linear propagated waves. Similar emergent patterns of diverse spatiotemporal excitations have been proposed for the human uterus using FitzHugh–Nagumo modelling (Aslanidi et al., 2011; Sheldon et al., 2013). Of note, simultaneous recordings of myometrial tissue membrane potential and force indicate two important features to consider here: there is a temporal lag of coupling between excitation and force development and, during action potentials with repetitive spiking, periods of brief repolarisations during which force is maintained (Wray et al., 2015). Thus, it is not necessary for complete spatiotemporal synchronisation of electrical activity to occur across the whole uterine tissue to support contractile force.

### Conclusions and future perspectives

In the case of human myometrium, evidence for a morphologically distinct pacemaking region(s) is lacking. The balance of opinion, at the moment, suggests for other species too that there are multiple uterine areas with the potential for initiating spontaneous action potential propagation. The proposal that emergence of co-ordinated electrical-contraction coupling resulting in spontaneous uterine contractions during labour can arise from spatiotemporally discrete areas of activity has several attractions. It provides a rationale for how spatially and temporally discrete electrical activities, in what is a non-homogeneous biological matrix, can become 'in-phase' sufficiently to co-ordinate contractile activity across the large uterine organ. Importantly, it does not preclude influences upon this arising from different intra-uterine physiological circumstances such as heterocellular interactions, paracrine stimulation within the maternal-placental interface or long-distance mechano-electrical feedback across fasciculi/bundles of the outer myometrium (Yochum et al., 2018; Young & Barendse, 2014). These are consistent with the suggestion that uterine contractions of labour arise as a result of the modular accumulation of physiological systems (Mitchell & Taggart, 2009).

There remain, however, many gaps in our knowledge of how uterine electrical activity originates and spreads and the manner(s) by which electrical-contraction coupling is regulated in frequency and amplitude across the organ. This much is evident by comparing to our detailed understanding of pacemaking in other physiological systems such as the heart where a depth of physiological knowledge underpins clinical therapies for altered pacemaking. This can be contrasted too with the unsatisfactory treatment options for spontaneous preterm birth, the rates of which have remained resistant to the clinical use of tocolytic drugs designed to reduce uterine contractions (Coler et al., 2021), or prolonged labours where scientific advances are not implemented in the clinic (Maaløe et al., 2023).

There are several areas to be recommended for future research. Single cell RNA-sequencing has begun to elucidate molecular enrichment signatures for different uterine cell-types (Garcia-Flores et al., 2023; Vento-Tormo et al., 2018). Extending this in the human setting with single nuclear-RNA-Seq, spatial transcriptomics and spatial proteomics to biopsies from different uterine areas (e.g. lower *vs.* upper segments, sub-basalis *vs.* sub-parietalis decidua in humans) holds promise for the geometric mapping of molecular characteristics that may indicate distinct cell types, or sub-types of cell, involved in pacemaking and electrical propagation. The utilisation of advancing techniques for measuring electrical activity *ex vivo* (e.g. optical mapping in conjunction with multi-electrode arrays) with continued improvement in spatiotemporal resolution over large tissue arrays is warranted. So too is necessary to pursue improvements for *in vivo* recordings (e.g. surface EHG recordings with electrodes minimally spaced across a large abdominal surface area in combination with magnetic imaging (EMMI, Wang et al., 2023) and algorithms for more refined uterine electrical signal deconvolution (Xu et al., 2022) offering less inconvenience, and thus, greater patient acceptability and as well as cost effectiveness. These approaches would facilitate the evolution of more detailed, experimentally informed quantitative models of single cell, tissue and organ electrical stimulation. In turn, the integration of all such data, while being a considerable challenge, will be necessary to inform the development of new therapeutic strategies to prevent uterine contractions of spontaneous preterm birth and reduce prolonged dysfunctional labours at term.

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

## Additional information

### Competing interests

The authors declare no conflict of interest.

### Author contributions

Both authors contributed to the conception or design of the work; drafting the work or revising it critically for important intellectual content; final approval of the version to be published; agreement to be accountable for all aspects of the work. All persons designated as authors qualify for authorship, and all those who qualify for authorship are listed.

### Funding

Support for research relevant to this review is acknowledged by MJT from MRC (G0902091 and MR/L0009560/1), Action Medical Research (GN2807) and Borne (BORNE-2021-0005).

### Acknowledgements

We appreciate the contribution of Dr W. C. Tong (Newcastle University) to Fig. 4.

### Keywords

action potential, excitability, imaging, labour, uterus

### Supporting information

Additional supporting information can be found online in the Supporting Information section at the end of the HTML view of the article. Supporting information files available:

**Peer Review History**

