## [Peer Review History · The Journal of Physiology]

An Update on Pacemaking in the myometrium

Susan Wray and Michael J Taggart

DOI: 10.1113/JP284753

Corresponding author(s): Susan Wray (s.wray@liv.ac.uk)

The following individual(s) involved in review of this submission have agreed to reveal their identity: Sean Marcus Ward (Referee #2)

Review Timeline:

Submission Date:	11-Dec-2023
Editorial Decision:	05-Jan-2024
Revision Received:	11-Apr-2024
Accepted:	24-Apr-2024

Senior Editor: Laura Bennet

Reviewing Editor: Bernard Drumm

Transaction Report:

Dear Professor Wray,

Re: JP-TR-2023-284753 "An Update on Pacemaking in the myometrium" by Susan Wray and Michael J Taggart

Thank you for submitting your manuscript to The Journal of Physiology. It has been assessed by a Reviewing Editor and by 2 expert referees and we are pleased to tell you that it is acceptable for publication following satisfactory revision.

ABSTRACT FIGURES: Authors may use The Journal's premium BioRender account to create/redraw their Abstract Figures (and any other suitable schematic figure). Information on how to access this account is here: <https://physoc.onlinelibrary.wiley.com/journal/14697793/biorender-access>.

REVISION CHECKLIST: Upload a full Response to Referees file. To create your 'Response to Referees' copy all the reports, including any comments from the Senior and Reviewing Editors, into a Microsoft Word, or similar, file and respond to each point, using font or background colour to distinguish comments and responses and upload as the required file type.

We look forward to receiving your revised submission.

Yours sincerely,

Professor Laura Bennet

Senior Editor
The Journal of Physiology
<https://jp.msubmit.net>
<http://jp.physoc.org>
The Physiological Society
Hodgkin Huxley House
30 Farringdon Lane
London, EC1R 3AW
UK
<http://www.physoc.org>
<http://journals.physoc.org>

EDITOR COMMENTS

Reviewing Editor:

Both referees have commented that this is a well-written timely review on uterine pacemaking. Both have highlighted some minor typos and grammatical errors that should be adjusted.

Both reviewer 1 and 2 have suggested some additions to the review to expand on some of the postulations which would make the review potentially more impactful. For example, we agree with Reviewer 1's suggestions of including an original summary diagram, as well as a novel figure showing the different action potential characteristics from the uterus of different species.

Reviewer 2 makes the suggestion that the authors could perhaps make the comparison between the uterus and other visceral organs when discussing pacemaking, with specific reference to the GI tract and the role of interstitial cells therein. In the original article, the authors did in fact do this in section 2.2, however an expanded paragraph in this section along the lines of the suggestions of reviewer 2 would be a valuable addition.

REFEREE COMMENTS

Referee #1:

The manuscript under consideration is a commissioned review from Susan Wray and Michael Taggart, both well-established experts and long-time contributors to the field of myometrial physiology and pacemaking. This reviewer enjoyed reading this work and found it to be extremely well-written with a logical flow. It will be a valuable resource to other investigators in the field. I suggest some minor additions as described below:

- What is known about uterine repair after injury due to eg. c-section. Does that affect pacemaker activity? Is it known? Remodeling of electrical behavior and scarring seems likely to affect propagation of action potentials. This may make for an interesting discussion point that could be added to the future perspectives section. Alternatively, it could be raised earlier in the review where damage from placenta removal is mentioned.
- The reader would benefit from the addition of a figure showing the typical myometrial cell action potentials from different animals would be a nice addition.
- An original summary figure would be a welcome addition.

Minor

1. Line 22 in the Abstract: remove the extra comma.
2. Rather than Figure 1 being "borrowed" from a 2022 book chapter by another group, I would encourage the authors to generate their own version. Each of the three figures are borrowed from previous publications, this seems one figure that could be generates afresh.
3. Line 80: "ae" should be "are"
4. Throughout there are tracked changes that need to be removed see lines 240, 254, 304-305, 309
5. Figure 3 is labeled in the legend as Fig. XX.

Referee #2:

This is a timely short review examining the current knowledge of pacemaker activity in the myometrium. This pacemaker activity underlies the periodic contractions of the uterine smooth muscle that are critical for successful pregnancy outcome. The review describes that the current knowledge in this area is lacking and that there are many gaps in our knowledge of this critical function of the uterus. The review examines recent technologies including in vivo and ex vivo imaging along with electrophysiology and computational modelling of both animal and human myometrial studies. It starts with a description of the uterus in different species, followed by an examination of functional studies investigating pacemaker mechanisms and uterine myocytes. The review concludes that there is no fixed pacemaking site in the human myometrium, but rather it is mobile and emphasizes that there should be more studies and funding to better understand pacemaker activity of the myometrium leading to better treatments for clinical conditions such as preterm and dysfunctional labors.

The reviewer has few criticisms of this review. One was the absence of a comparison with pacemaker mechanisms in other visceral organs rather than the comparison with the nodal region of the heart. Considerable evidence has emerged over the last number of years demonstrating the role of interstitial cells of Cajal as pacemakers in gastrointestinal muscles. Although dominant pacemaker activity exists from these cells that usually lie in a distinct intramuscular plane in several organs of rodents (i.e., mice, stomach and small intestine), the dominant site of pacemaker activity within these networks can change during a recording period, either using microelectrodes or more recently cell-specific imaging techniques. This may be like that reported in the uterus that the authors describe in the current article. It may be worthwhile to add a paragraph to the article describing these similarities.

A few minor points are identified that should be corrected.

Line 80: "are to be produced"

Line 240: red line.

Line 253: ((.

Line 260: (see (Parkington.

Line 303: red line through reference.

Line 309: red).

Line 327: ((.

Line 339: (Malone...

Line 342: ((.

Line 353:))).

REQUIRED ITEMS

- Please include an Abstract Figure file, as well as the Figure Legend text within the main article file. The Abstract Figure is a piece of artwork designed to give readers an immediate understanding of the Review Article and should summarise the main conclusions. If possible, the image should be easily 'readable' from left to right or top to bottom. It should show the physiological relevance of the Review so readers can assess the importance and content of the article. Abstract Figures should not merely recapitulate other figures in the Review. Please try to keep the diagram as simple as possible and without superfluous information that may distract from the main conclusion of the Review. Abstract Figures must be provided by authors no later than the revised manuscript stage and should be uploaded as a separate file during online submission labelled as File Type 'Abstract Figure'. Please ensure that you include the figure legend in the main article file. All Abstract Figures will be sent to a professional illustrator for redrawing and you may be asked to approve the redrawn figure before your paper is accepted.

- Your MS must include a complete "Additional information section" with the following 4 headings and content:

Competing Interests: A statement regarding competing interests. If there are no competing interests, a statement to this effect must be included. All authors should disclose any conflict of interest in accordance with journal policy.

Author contributions: Each author should take responsibility for a particular section of the study and have contributed to writing the paper. Acquisition of funding, administrative support or the collection of data alone does not justify authorship; these contributions to the study should be listed in the Acknowledgements. Additional information such as 'X and Y have contributed equally to this work' may be added as a footnote on the title page.

It must be stated that all authors approved the final version of the manuscript and that all persons designated as authors qualify for authorship, and all those who qualify for authorship are listed.

Funding: Authors must indicate all sources of funding, including grant numbers. If authors have not received funding, this must be stated.

It is the responsibility of authors funded by RCUK to adhere to their policy regarding funding sources and underlying research material. The policy requires funding information to be included within the acknowledgement section of a paper. Guidance on how to acknowledge funding information is provided by the Research Information Network. The policy also requires all research papers, if applicable, to include a statement on how any underlying research materials, such as data, samples or models, can be accessed. However, the policy does not require that the data must be made open. If there are considered to be good or compelling reasons to protect access to the data, for example commercial confidentiality or legitimate sensitivities around data derived from potentially identifiable human participants, these should be included in the statement.

Acknowledgements: Acknowledgements should be the minimum consistent with courtesy. The wording of acknowledgements of scientific assistance or advice must have been seen and approved by the persons concerned. This section should not include details of funding.

- Please upload separate high quality figure files via the submission form.

- Author profile(s) must be uploaded via the submission form. Authors should submit a short biography (no more than 100 words for one author or 150 words in total for two authors) and a portrait photograph of the two leading authors on the paper. These should be uploaded and clearly labelled together in a Word document with the revised version of the manuscript. Any standard image format for the photograph is acceptable, but the resolution should be at least 300 DPI and preferably more. A group photograph of all authors is also acceptable, providing the biography for the whole group does not exceed 150 words.

- It is the authors' responsibility to obtain any necessary permissions to reproduce previously published material and to list these within the main article file. For information, please see: https://jp.msubmit.net/cgi-bin/main.plex?form_type=display_requirements#permissions.

END OF COMMENTS

Confidential Review

11-Dec-2023

We thank the reviewing editor and referee's for their considered comments. Our responses are outlined below.

Reviewing Editor:

Both referees have commented that this is a well-written timely review on uterine pacemaking. Both have highlighted some minor typos and grammatical errors that should be adjusted.

THANK YOU. REVISIONS FOR TYPOS MADE.

Both reviewer 1 and 2 have suggested some additions to the review to expand on some of the postulations which would make the review potentially more impactful. For example, we agree with Reviewer 1's suggestions of including an original summary diagram, as well as a novel figure showing the different action potential characteristics from the uterus of different species.

THESE FIGURES HAVE BEEN MADE.

Reviewer 2 makes the suggestion that the authors could perhaps make the comparison between the uterus and other visceral organs when discussing pacemaking, with specific reference to the GI tract and the role of interstitial cells therein. In the original article, the authors did in fact do this in section 2.2, however an expanded paragraph in this section along the lines of the suggestions of reviewer 2 would be a valuable addition.

WE TOOK FURTHER ADVICE FROM THE EDITORS ON THIS POINT, SO AS NOT TO OVERLAP WITH EXPERT TEXT FROM AUTHORS ALSO PUBLISHING ARTICLES IN THE SPECIAL ISSUE. WE HAVE ADDED SOME ADDITIONAL TEXT, DETAILED IN OUR RESPONSES TO REVIEWER 2.

REFeree COMMENTS

Referee #1:

The manuscript under consideration is a commissioned review from Susan Wray and Michael Taggart, both well-established experts and long-time contributors to the field of myometrial physiology and pacemaking. This reviewer enjoyed reading this work and found it to be extremely well-written with a logical flow. It will be a valuable resource to other investigators in the field. I suggest some minor additions as described below:

- What is known about uterine repair after injury due to eg. c-section. Does that affect pacemaker activity? Is it known? Remodeling of electrical behavior and scarring seems likely to affect propagation of action potentials. This may make for an interesting discussion point that could be added to the future perspectives section. Alternatively, it could be raised earlier in the review where damage from placenta removal is mentioned.

We thank the reviewer for this interesting comment. There are no direct data on this question that we can find. By analogy with damaged cardiac tissue and arrhythmias, it could be speculated that C-section scar tissue, which has more collagen than normal uterine tissue, may lead to abnormal

electrical activity. The only study we could find was comparing the subendometrial contractility in non-pregnant women with and without a surgical scar. There was a finding of less co-ordinated activity as detected by ultrasound speckle tracking. Whether this is due to altered pacemaking is hard to assess.

There is increasing evidence that emergency C-section late in pregnancy imparts a greater risk of spontaneous preterm birth in each subsequent pregnancy. The focus of attention on underlying reasons has been on damage to cervical tissue due to the site of low surgical incision and that this may lead to cervical incompetence in subsequent pregnancies. Quite where the boundary of uterine and cervical (smooth muscle) tissue lies is uncertain. If there is a continuum of smooth muscle tissue it may also be possible that electrical conductivity will be disrupted but, as above, this has yet to be not investigated.

To reflect the reviewer's suggestion, we have referred to these studies with some speculation at the end of the paragraph on subendometrial activity on page 6.

- The reader would benefit from the addition of a figure showing the typical myometrial cell action potentials from different animals would be a nice addition.

DONE. This is now Figure 3 in the revised manuscript.

- An original summary figure would be a welcome addition.

DONE

Minor

1. Line 22 in the Abstract: remove the extra comma. **CORRECTED**

2. Rather than Figure 1 being "borrowed" from a 2022 book chapter by another group, I would encourage the authors to generate their own version. Each of the three figures are borrowed from previous publications, this seems one figure that could be generated afresh.

DONE

3. Line 80: "ae" should be "are" **CORRECTED**

4. Throughout there are tracked changes that need to be removed see lines 240, 254, 304-305, 309 **CORRECTED**

5. Figure 3 is labeled in the legend as Fig. XX. **CORRECTED**

Referee #2:

This is a timely short review examining the current knowledge of pacemaker activity in the myometrium. This pacemaker activity underlies the periodic contractions of the uterine smooth muscle that are critical for successful pregnancy outcome. The review describes that the current knowledge in this area is lacking and that there are many gaps in our knowledge of this critical

function of the uterus. The review examines recent technologies including in vivo and ex vivo imaging along with electrophysiology and computational modelling of both animal and human myometrial studies. It starts with a description of the uterus in different species, followed by an examination of functional studies investigating pacemaker mechanisms and uterine myocytes. The review concludes that there is no fixed pacemaking site in the human myometrium, but rather it is mobile and emphasizes that there should be more studies and funding to better understand pacemaker activity of the myometrium leading to better treatments for clinical conditions such as preterm and dysfunctional labors.

The reviewer has few criticisms of this review. One was the absence of a comparison with pacemaker mechanisms in other visceral organs rather than the comparison with the nodal region of the heart. Considerable evidence has emerged over the last number of years demonstrating the role of interstitial cells of Cajal as pacemakers in gastrointestinal muscles. Although dominant pacemaker activity exists from these cells that usually lie in a distinct intramuscular plane in several organs of rodents (i.e., mice, stomach and small intestine), the dominant site of pacemaker activity within these networks can change during a recording period, either using microelectrodes or more recently cell-specific imaging techniques. This may be like that reported in the uterus that the authors describe in the current article. It may be worthwhile to add a paragraph to the article describing these similarities.

THANK YOU FOR THIS POINT. WE HAVE BALANCED BETWEEN FOCUSING ON WHAT IS UNIQUE TO THE MYOMETRIUM AND REFERENCING BRIEFLY OTHER MAJOR PACEMAKING MECHANISMS. WE HAVE DONE THIS IN THE LIGHT THAT OTHER PAPERS WILL ADDRESS THESE OTHER SMOOTH MUSCLE PACEMAKERS, IN THE SPECIAL ISSUE.

WE DO HOWEVER TAKE THE REVIEWER'S POINT AND HAVE DRAWN ATTENTION TO THE INTERESTING POINT ABOUT MOBILITY OF PACEMAKERS AND CITE ONE OF THE EARLIEST STUDIES OF THIS BY BAUER, PUBLICOVER AND SAUNDERS FROM 1985. WE HAVE ADDED TEXT TO P9

"In other papers associated with this Special Issue, the nature of pacemaking in the gastrointestinal tract (GIT) is described. The study of the origins of slow waves in the GIT has been greatly aided by the fine anatomical details of these smooth muscles tissues and the identification of specialized cells, such as ICCs. It is worth noting that when electrical mapping is undertaken, the pacemaking site is seen to shift, "from event to event, even within a small population of muscle cells". Descriptions such as these points to some similarity with findings in the uterus. However, unlike the uterus, in the GIT the origin of the pacemaker was not random, and occurred in hotspots, within certain areas of the tissue (Bauer et al., 1985). For a more detailed account, recent papers from the authors can be consulted, along with several comprehensive reviews (Huizinga, 2019; Parsons & Huizinga, 2018; Sanders, 2019)."

A few minor points are identified that should be corrected.

Line 80: "are to be produced" **CORRECTED**

Line 240: red line. **CORRECTED**

Line 253: ((. **CORRECTED**

Line 260: (see (Parkington. **CORRECTED**

Line 303: red line through reference. **CORRECTED**

Line 309: red). **CORRECTED**

Line 327: ((. **CORRECTED**

Line 339: (Malone... **CORRECTED**

Line 342: ((. **CORRECTED**

Line 353:)). **CORRECTED**

Dear Professor Wray,

Re: JP-TR-2024-284753R1 "An Update on Pacemaking in the myometrium" by Susan Wray and Michael J Taggart

We are pleased to tell you that your paper has been accepted for publication in The Journal of Physiology.

Authors should note that it is too late at this point to offer corrections prior to proofing. Major corrections at proof stage, such as changes to figures, will be referred to the Editors for approval before they can be incorporated. Only minor changes, such as to style and consistency, should be made at proof stage. Changes that need to be made after proof stage will usually require a formal correction notice.

Yours sincerely,

Laura Bennet
Senior Editor
The Journal of Physiology

P.S. - You can help your research get the attention it deserves! Check out Wiley's free Promotion Guide for best-practice recommendations for promoting your work at www.wileyauthors.com/eoo/guide. You can learn more about Wiley Editing Services which offers professional video, design, and writing services to create shareable video abstracts, infographics, conference posters, lay summaries, and research news stories for your research at www.wileyauthors.com/eoo/promotion.

IMPORTANT NOTICE ABOUT OPEN ACCESS: To assist authors whose funding agencies mandate public access to published research findings sooner than 12 months after publication, The Journal of Physiology allows authors to pay an Open Access (OA) fee to have their papers made freely available immediately on publication.

You can check if your funder or institution has a Wiley Open Access Account here: <https://authorservices.wiley.com/author-resources/Journal-Authors/licensing-and-open-access/open-access/author-compliance-tool.html>.

EDITOR COMMENTS

Reviewing Editor:

Thank you for comprehensively addressing the reviewer comments and for adding to your review based on their feedback. As noted by both reviewers, this is a timely review on uterine pacemaking.